# STAR : Semantic–ID Token-Embedding Alignment for Generative Recommenders

## Abstract

Generative recommenders (GRs)—which directly generate the next-item semantic ID with an autoregressive model—are rapidly gaining adoption in research and large-scale production as a scalable, efficient alternative to traditional recommendation algorithms. Yet we find a degenerate solution when adapting Language Models (LMs) to GRs. We identify, for the first time, a pervasive token–embedding misalignment issue: the common mean-of-vocabulary initialization places new Semantic-ID tokens on the LM manifold but collapses their distinctions, stripping item-level semantics and degrading data efficiency and retrieval quality. We introduce **STAR**, a lightweight alignment stage that freezes the LM and updates *only* Semantic-ID embeddings via paired supervision from item titles/descriptions $\leftrightarrow$ Semantic-ID, thereby injecting the new tokens with linguistically grounded, item-level semantics while preserving the pretrained model's capabilities and the primary recommendation objective. Across multiple datasets and strong baselines, **STAR** consistently improves top-$k$ retrieval/search performance over mean-of-vocabulary initialization and status-quo auxiliary-task adaptation. Ablations and analyses corroborate our claims, showing increased token-level diversity, stronger linguistic grounding, and improved sample efficiency. **STAR** is parameter-efficient, updating only the Semantic-ID token embeddings ($|\mathcal{V}_{\text{SemID}}| \times D$ parameters), and integrates seamlessly with standard GR pipelines.

## 1 Introduction

Generative Recommenders (GRs) (Rajput et al., 2023; Deldjoo et al., 2024; Zhai et al., 2024) have emerged as a promising paradigm in recommendation systems, attracting increasing attention in both academia and industry . Traditional embedding-based approaches—such as matrix factorization (MF) (Koren et al., 2009), neural collaborative filtering (NCF) (He et al., 2017), LightGCN (He et al., 2020), and NGCF (Wang et al., 2019)—suffer from fundamental computational constraints: scoring requires dense user–item inner products for large candidate sets, leading to prohibitive inference cost or substantial memory overhead for approximate indexing. In contrast, GRs address these limitations via two key innovations: (i) they employ autoregressive modeling to encode user preferences directly from the interaction history (Zhai et al., 2024), and (ii) they generate recommendations token-by-token without explicit user–item dot-product computation. Moreover, by building on autoregressive architectures, GRs can exploit established scaling-law behavior (Han et al., 2025)—achieving predictable quality gains as model size, data, and compute increase—thereby offering a clear path to continued performance improvement as resources scale.

Generative recommenders typically use a two-stage pipeline: an RQ-VAE maps items to semantic IDs, and a transformer autoregressively predicts the next ID from a user's history (Lee et al., 2022). This pipeline has significantly advanced the field. However, it follows two typical paradigms, each of which ultimately strips away item-level semantics and undermines both data efficiency and retrieval quality: **Standard sequential approaches** trained from scratch to model next-token probabilities but without explicitly capturing the semantic meaning of Semantic IDs—leading to lower data efficiency and weaker retrieval due to reliance on collaborative signals alone (Zheng et al., 2024). **Language model adaptation approaches**, conversely, leverage pre-trained large language models to interpret semantic IDs (Zheng et al., 2024; Chen et al., 2025a), demonstrating that integrating linguistic and collaborative semantics yields substantial performance improvements. Nevertheless, the auxiliary

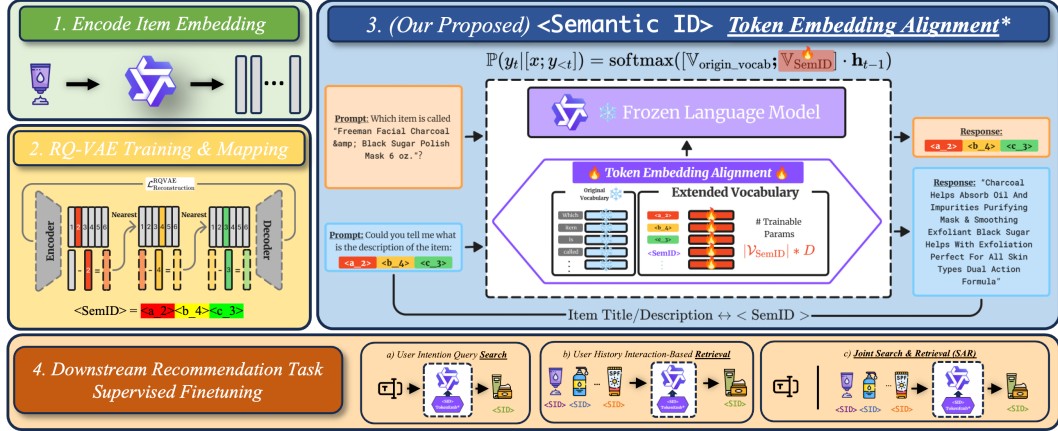

Figure 1: Overview of our proposed **STAR**. Items are encoded into dense vectors and discretized with a residual-quantized VAE (RQ-VAE) to produce Semantic-ID sequences (e.g., $\langle a_2 \rangle \langle b_4 \rangle \langle c_3 \rangle$). **Our key contribution is a lightweight *token-embedding alignment* method** that extends a *frozen* language model with the SemID vocabulary and aligns the new Semantic-ID token embeddings to the LM's embedding space via item titles/descriptions $\leftrightarrow$ SemID supervision, training only $|\mathcal{V}_{\text{SemID}}| \times D$ parameters. This resolves Semantic-ID token embedding misalignment problem. The aligned model is then supervised-fine-tuned on downstream recommendation tasks, yielding improved end-task (retrieval and search) performance.

tasks proposed by (Zheng et al., 2024) require the model to unnecessarily memorize all items, which may impede retrieval performance.

In this paper, we identified these difficulties by uncovering a fundamental limitation which we define as **Semantic IDs token–embedding misalignment**. In standard sequential setups, randomly initialized models (e.g., T5-Small used as a sequence model) treat semantic IDs as arbitrary tokens with no inherent semantics—lacking both world knowledge and linguistic structure (Rajput et al., 2023). Consequently, such models **require large amounts of training data to merely learn collaborative semantics (co-occurrence patterns), rather than to exploit any linguistic meaning of the IDs themselves**. Previous work on Language-model adaptation (Wu et al., 2024) exhibits a related failure mode: semantic IDs correspond to new, out-of-vocabulary tokens whose embeddings are typically randomly initialized or heuristically averaged from existing tokens, leaving them misaligned with the pretrained embedding space at initialization. Prior solution (Zheng et al., 2024) attempts to inject linguistic information via carefully designed auxiliary tasks, but this multi-task formulation introduces an objective mismatch: **auxiliary losses that encourage memorization are not tightly coupled to the primary next-item retrieval objective in sequential recommendation**, yielding inconsistent gains across datasets and evaluation protocols.

To avoid this pitfall, we do not simply rely on random (or mean-of-vocabulary) initialization or coarse auxiliary objectives that loosely pull Semantic-ID embeddings toward proxy linguistic signals. Instead, we frame token embedding misalignment as a principled embedding adaptation problem: ensuring that newly introduced tokens inherit meaningful, linguistically grounded representations while remaining compatible with the pretrained LM's embedding space. The key idea is to directly endow Semantic-ID embeddings with item-level semantics derived from content supervision, thereby resolving the mismatch between well-trained vocabulary embeddings and newly initialized identifiers. This perspective shifts the focus from ad-hoc heuristics to a targeted alignment stage that preserves the LM's pretrained geometry while improving downstream retrieval performance.

We introduce **STAR**, a lightweight token-embedding alignment method that addresses the token-embedding misalignment identified above. **STAR** learns token embeddings for newly introduced Semantic-ID tokens, grounding their linguistic semantics and aligning them with the pretrained LM's token-embedding space. This resolves the mismatch between well-trained vocabulary embeddings and newly initialized Semantic-ID embeddings, yielding consistent gains on recommendation tasks.

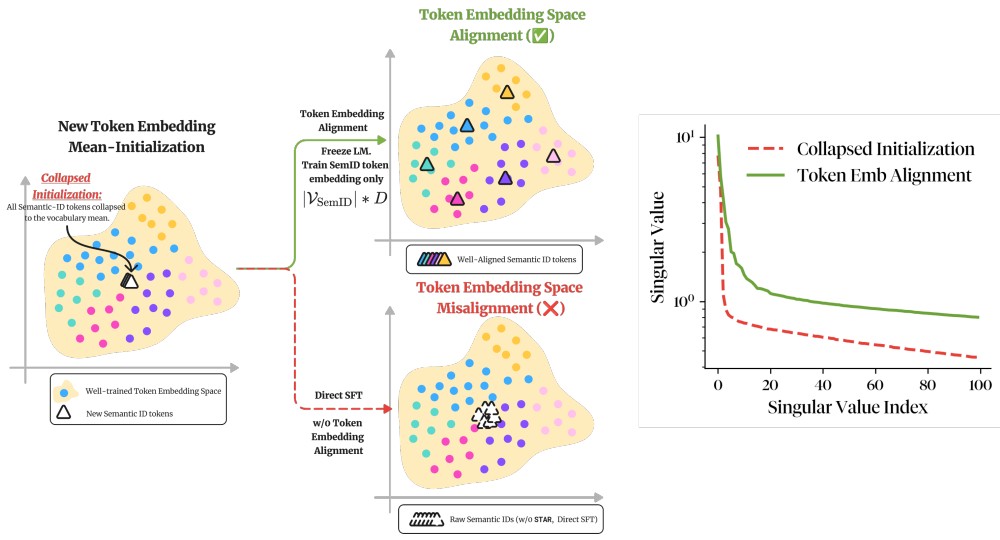

Figure 2: *Left:* **Token embedding-space misalignment and alignment of Semantic-ID tokens.** Newly introduced SemID embeddings (white triangles) are initialized at the vocabulary mean, producing an *in-manifold* yet semantically uninformative collapse within the pretrained LM's embedding space. With the LM frozen, we update only the SemID rows ($|\mathcal{V}_{\text{SemID}}| \times D$) via item-title/description $\leftrightarrow$ SemID supervision, dispersing them into semantically consistent neighborhoods on the LM manifold and enabling effective downstream fine-tuning; without this alignment stage, the SemIDs remain collapsed. *Right:* **Singular-value spectrum of Semantic-ID embeddings after identical supervised fine-tuning.** Stacking the learned Semantic-ID vectors into a matrix $E \in \mathbb{R}^{|\mathcal{V}_{\text{SemID}}| \times D}$, we plot the ordered singular values $\{\sigma_i(E)\}$ (log scale). Token-embedding alignment yields a slower spectral decay and higher effective rank than mean-of-vocabulary ("collapsed") initialization, indicating greater item-level diversity and a non-degenerate Semantic-ID subspace.

**Contributions.** Our work makes three contributions:

1. First, we **identify and define the Semantic IDs token embedding misalignment problem** that arises when integrating semantic ID tokens into existing language model architectures, providing empirical evidence through extensive experiments.
2. Second, we introduce **a lightweight token-embedding alignment method, called `STAR`**, that effectively aligns semantic ID token embeddings with established vocabulary token embedding space while preserving their linguistic semantic properties without compromising the primary retrieval objective.
3. Third, through **comprehensive evaluations on diverse datasets and tasks**, we show that `STAR` delivers robust, consistent gains in both sequential recommendation and query-to-item retrieval, outperforming strong baselines.

Overall, this work diagnoses and remedies the misalignment issue of semantic ID token embeddings which significantly hinders the downstream recommendation task performance and provides a principled, lightweight solution: pretraining semantic-ID token embeddings to be linguistically grounded and aligned with the well-pretrained LM's token embedding space *prior to* downstream supervised finetuning. This simple step improves data efficiency and end-task retrieval performance across datasets, offering a practical path for integrating semantic-ID tokens into pretrained LMs.

## 2 THE MISALIGNMENT ISSUE IN TOKEN EMBEDDING SPACE

In this section, we first formalize the generative retrieval setting. We then diagnose a systematic token-embedding misalignment introduced by standard language-model adaptation practices. Specifically, when new semantic-ID tokens are appended to the vocabulary of a pretrained language model, their embeddings are typically (i) randomly initialized or (ii) set to the mean of existing token embeddings.

These heuristics produce embeddings that are poorly aligned with the model's well-trained embedding space. Thus, the new tokens lack proper linguistic grounding, which in turn degrades supervised fine-tuning performance on downstream recommendation tasks.

## 2.1 PRELIMINARIES ON GENERATIVE RETRIEVAL

Our experiments follow the generative retrieval framework proposed by Rajput et al. (2023). For each item, we assume access to a set of content features (e.g., title, genres, description). These features are concatenated and then passed through a pre-trained encoder to obtain a semantic embedding $x \in \mathbb{R}^d$. The choice of encoder depends on the modality of the available content features. For instance, since our dataset contains only textual features, we employ a language model as the encoder. Formally, let $\mathcal{I}$ denote the set of items. Each item $I_i \in \mathcal{I}$ is associated with $p$ different content features $\{f_i^{(1)}, f_i^{(2)}, \ldots, f_i^{(p)}\}$. These features are concatenated into a prompt $P_i = \mathrm{Prompt}(f_i^{(1)}, f_i^{(2)}, \ldots, f_i^{(p)})$, and then mapped to an embedding space by a modality-specific encoder

$$\mathrm{Enc} : \mathcal{X} \to \mathbb{R}^d, \text{with } z_i = \mathrm{Enc}(P_i)$$

In our setting, $\mathcal{X}$ denotes the text space and $\mathrm{Enc}()$ denotes a well-trained language embedding model. After obtaining the semantic embeddings, we convert them into discrete semantic IDs using the Residual Quantized Variational Autoencoder (RQ-VAE). Formally, given an embedding $z \in \mathbb{R}^d$, we initialize the residual as $r_0 := z$. At each level $l \in \{0, \ldots, L-1\}$, with codebook $\mathcal{C}_d = \{e_k^{(l)}\}_{k=1}^K$, we compute $c_l = \arg\min_k \|r_l - e_k^{(l)}\|$, $r_{l+1} := r_l - e_{c_l}^{(l)}$. This recursive process yields a tuple of indices $(c_0, \ldots, c_{L-1})$, which defines the semantic ID of $z$. The quantized representation is then $\hat{z} = \sum_{d=0}^{L-1} e_{c_l}^{(l)}$, which is passed to the decoder for reconstruction.

The RQ-VAE performs coarse-to-fine quantization: early levels capture coarse information, while later levels refine smaller residuals. Training jointly optimizes the encoder, decoder, and codebooks by minimizing the reconstruction loss with a regularization term enforcing codebook commitment. Thus, the RQ-VAE establishes a mapping from the continuous embedding space to the discrete semantic ID space:

$$\phi : \mathbb{R}^d \to \mathcal{C}^L, \quad \phi(z) = (c_0, \ldots, c_{L-1})$$

To adapt a pretrained language model (LM) for generative recommendation, these newly introduced semantic IDs must be integrated into the LM's token vocabulary. Following standard practice (Hewitt, 2021), the embedding vectors of the new tokens are initialized as the mean of the pretrained vocabulary embeddings[1]:

$$\mathbf{e}_{\mathrm{tokemb}}^{<c_i>} := \frac{1}{|\mathcal{V}_{\mathrm{text}}|} \sum_{v \in \mathcal{V}_{\mathrm{text}}} \mathbf{e}_v \tag{1}$$

Finally, we convert a user's interaction history into a sequence of discrete semantic IDs and model it autoregressively with a transformer. Concretely, we concatenate the sequence of discrete semantic IDs as $c_0^1, c_1^1, ..., c_{L-1}^1, c_0^2, c_1^2, ..., c_{L-1}^2, c_0^n, c_1^n, ..., c_{L-1}^n$, and the generative retrieval objective is defined as $P(c_1, c_2, ..., c_T) = \prod_{t=1}^T P(c_t | c_{<t})$.

In this formulation, the interaction-history–based retrieval and recommendation task is recast as a sequential generative retrieval (GR) problem. By modeling the sequence of discrete semantic IDs autoregressively, we leverage the powerful sequential modeling capabilities of transformer architectures. This approach has been shown to achieve both high effectiveness and computational efficiency (Yang et al., 2024), making it well-suited for large-scale recommendation scenarios.

## 2.2 DIAGNOSTICS OF TOKEN EMBEDDING MISALIGNMENT

Given the problem setup, a critical challenge arises when adapting pretrained LMs: the initialization of Semantic-ID token embeddings. We show that the common mean-of-vocabulary initialization yields a degenerate solution, leading to systematic token-embedding misalignment. The prevailing approach extends the pretrained vocabulary with newly introduced Semantic-ID tokens and initializes their embeddings to the mean of the existing token embeddings (Hewitt, 2021), as shown in

---

[1] $\mathbf{e}_{\mathrm{tokemb}}^{\langle c_i \rangle}$ denotes the token embedding corresponding to the Semantic ID $\langle c_i \rangle$ in the language model.

equation 1. While this mean-of-vocabulary scheme places the new tokens on the pretrained embedding manifold—and can yield a tighter KL divergence upper bound for their probabilities—it collapses them into an undifferentiated region, erasing item-level distinctions (Fig. 2). Contrary to the intent of "good" initialization—facilitating rapid adaptation to the downstream domain—this practice fails to exploit the latent linguistic structure associated with Semantic IDs, thereby hindering downstream recommendation performance. We empirically show that introducing an explicit token embedding alignment stage to endow Semantic-ID tokens with linguistically grounded, item-level semantics substantially improves generalization in downstream retrieval recommendation and search recommendation.

## 3 **STAR**: OUR PROPOSED TOKEN EMBEDDING PRETRAINING

In this section, we introduce an innovative method, termed **STAR**, designed to address the Semantic-ID token embedding misalignment problem outlined above. The objective of **STAR** is to enable a well-trained language model to effectively interpret newly incorporated Semantic-ID tokens prior to supervised fine-tuning on recommendation tasks, thereby improving both generalization and sample efficiency. We conduct extensive experiments across diverse datasets and competitive baselines, and further validate the approach on multiple recommendation scenarios, including **both retrieval and search tasks**. The results demonstrate that **STAR** delivers substantial performance gains, highlighting its effectiveness and broad applicability.

### 3.1 METHODOLOGY

We propose **STAR**, a lightweight *token-embedding alignment* stage that remedies the mismatch induced by injecting Semantic-ID tokens into a well-pretrained language model. The overall framework is presented in Fig. 1. Inspired by the vocabulary-extension insight of Toolken-style methods—namely, teaching a largely frozen LM to use newly added tokens by updating only their embeddings (Hao et al., 2024)—we adapt and specialize this idea to Generative Recommendations (GRs): we freeze all backbone parameters and update only the embeddings of the Semantic-ID vocabulary using a curated alignment corpus that pairs item titles/descriptions with their corresponding Semantic-ID sequences. This targeted alignment grounds the new tokens in the model's pretrained embedding manifold while keeping the backbone intact, mitigating initialization mismatch and improving sample efficiency and downstream task performance. After this stage, we follow standard language-model adaptation and perform supervised fine-tuning on downstream recommendation tasks.

Let $\mathcal{V} = \mathcal{V}_{\text{text}} \cup \mathcal{V}_{\text{SemID}}$ denote the extended vocabulary obtained by adding a set of Semantic-ID tokens $\mathcal{V}_{\text{SemID}}$ to a well-pretrained autoregressive language model (LM) with parameters $\theta$ and input-embedding matrix $E \in \mathbb{R}^{|\mathcal{V}| \times d}$. We write $E_{\text{SemID}} \in \mathbb{R}^{|\mathcal{V}_{\text{SemID}}| \times d}$ for the rows of $E$ associated with Semantic-ID tokens and $E_{\text{text}}$ for the remainder. Each item $I_i$ is represented as a short natural-language description $f_i$ (e.g., title/description) and a canonical Semantic-ID sequence $y_i = (c_{i,0}, \ldots, c_{i,m-1})$ used by generative recommenders (GRs) for next-item generation. The standard adaptation pipeline initializes $E_{\text{SemID}}$ via mean-of-vocabulary or random schemes and directly proceeds to supervised fine-tuning (SFT) on interaction data, which produces a persistent embedding mismatch (§2.2).

**Goal.** We seek to *align* the newly introduced Semantic-ID embeddings to the well-pretrained token manifold *before* SFT, so that (i) the LM can already "speak" the Semantic-ID vocabulary from textual descriptions, and (ii) downstream SFT can focus on historical interaction modeling rather than repairing poor token initialization.

**Token-Embedding Alignment.** We introduce **STAR**, a lightweight stage that freezes all backbone parameters and updates only $E_{\text{SemID}}$ using a curated text↔Semantic-ID alignment corpus. Conceptually, **STAR** specializes the vocabulary-extension insight from Toolken-style methods—teaching a largely frozen LM to use newly added tokens by training their embeddings (Hao et al., 2024; Nguyen et al., 2024)—to the generative-recommendation (GR) setting, where the new tokens denote *items*

rather than tools and must encode fine-grained lexical semantics[2]. First, we construct an alignment dataset $\mathcal{D}_{\text{align}} = \{(x_i, y_i)\}_{i=1}^n$, where $x_i$ is an item's title/description and $y_i$ is its Semantic-ID sequence [3]. We use an instruction-style prompt template $\texttt{prompt}(x)$ to elicit generation of $y$ from $x$ (Template details in Appendix A.2). Then, we set our primary objective as a *generative* loss that conditions on the natural-language description and teacher-forces the Semantic-ID sequence:

$$\arg\min_{E_{\text{SemID}}} \sum_{(x,y)\in\mathcal{D}_{\text{align}}} \sum_{t=1}^{|y|} -\log \mathbb{P}_{[\theta; E_{\text{text}} \cup E_{\text{SemID}}]}\big(y_t \big| y_{<t}, \texttt{prompt}(x)\big), \tag{2}$$

With *all* LM parameters—including $E_{\text{text}}$—held fixed, we update only the corresponding rows of $E_{\text{SemID}}$. The loss grounds each Semantic-ID token's meaning in context and its interactions with natural-language tokens. After this alignment stage, we keep the learned Semantic-IDs as the initialization for downstream GR training and proceed with standard supervised fine-tuning on next-item generation, unfreezing model components as desired. Implementation details, see Algo 1.

## 3.2 Experiments

### 3.2.1 Setup

**Datasets.** To evaluate the effectiveness of our proposed **STAR**, we conduct extensive experiments on nine datasets covering diverse sources (Amazon (He & McAuley, 2016) and Yelp (Yelp, 2025)) and item categories. Specifically, we randomly pick four categories from the Amazon Product Reviews dataset for retrieval and search recommendation: `Arts`, `Beauty`, `Games`, and `Instruments`. As a fifth dataset, we use the `Yelp` Open Dataset (Yelp, 2025), which records user–business interactions on the Yelp platform. More Dataset details are provided in the Appendix A.1.

**Baselines.** We compare **STAR** against a broad set of competitive baselines, spanning traditional and LLM-based generative recommenders. *For Traditional Recommender*, we include: 1) **MF**: matrix factorization of user–item interactions (Koren et al., 2009). 2) **Caser**: CNN-based sequential model capturing local and positional patterns (Tang & Wang, 2018). 3) **HGN**: hierarchical GNN modeling high-order user–item connectivity (Ma et al., 2019). 4) **BERT4Rec**: bidirectional self-attention for sequential recommendation (Sun et al., 2019). 5) **LightGCN**: simplified GCN with linear message passing for high-order collaborative signals (He et al., 2020). 6) **SASRec**: employ self-attention mechanisms for long-range dependencies (Kang & McAuley, 2018a). *For LLM-based Generative Recommender*, we include: 1) **BIGRec**: LLM-based generative recommender using item titles as textual identifiers (Bao et al., 2023). 2) **P5-TID**: P5 variant treating item titles as textual IDs (Geng et al., 2023). 3) **P5-SemID**: constructs semantic identifiers from item metadata (e.g., attributes) (Geng et al., 2023). 4) **P5-CID**: injects collaborative signals via a spectral-clustering tree over item co-occurrence graphs (Geng et al., 2023). 5) **LC-Rec**: codebook-based identifiers (i.e. Semantic IDs) with auxiliary alignment objectives linking generated codes to natural language (Zheng et al., 2024).

**Evaluation Protocol and Metrics.** Following the standard evaluation protocol (Kang & McAuley, 2018b; Geng et al., 2023), we adopt a leave-one-out strategy for dataset splitting. Specifically, for each user sequence, the last interacted item is reserved for testing, the second-to-last item is used for validation, and the remaining items constitute the training set. We evaluate recommendation performance using Top-$K$ Recall (Recall@$K$) and Normalized Discounted Cumulative Gain (NDCG@$K$) with $K = 5, 10$ to evaluate the recommendation performance.

**Implementation Details.** We extract item-level semantic representations using the off-the-shelf `Qwen3-Embedding-0.6B` encoder, yielding 1024-dimensional vectors. For Semantic-ID tokenization, we follow Rajput et al. (2023): a 3-layer MLP encoder–decoder with ReLU activations and a 4-layer residual codebook (256 entries per layer, 32-dimensional codes). To encourage balanced codebook utlization, we add the diversity regularizer of Wang et al. (2024). The RQ-VAE is trained

---

[2]Unlike prior Toolken-style work, our goal is not to endow a model with tool-calling behaviors, but to provide a strong *vocabulary initialization* for Semantic-ID tokens that improves sample efficiency and downstream supervised fine-tuning for GR.

[3]We also include reversed pairs $\{(y_i, x_i)\}$ to enable bidirectional alignment.

Table 1: Overall performance comparison of sequential recommendation for both traditional and LLM-based algorithms. The last row shows **STAR**'s relative (%) gain over the status-quo competitive LLM-based baseline `LC-Rec`. Bold and underline are used to denote the best metric. Standard deviations over three independent trials are reported in (std).

| Methodology | Arts | | | | Games | | | | Yelp | | | |
|---|---|---|---|---|---|---|---|---|---|---|---|---|
| | R@5 | R@10 | N@5 | N@10 | R@5 | R@10 | N@5 | N@10 | R@5 | R@10 | N@5 | N@10 |
| MF | 0.0323 | 0.0486 | 0.0203 | 0.0266 | 0.0190 | 0.0340 | 0.0118 | 0.0167 | 0.0185 | 0.0292 | 0.0115 | 0.0149 |
| Caser | 0.0281 | 0.0421 | 0.0168 | 0.0213 | 0.0244 | 0.0418 | 0.0147 | 0.0203 | 0.0140 | 0.0239 | 0.0087 | 0.0119 |
| HGN | 0.0401 | 0.0545 | 0.0302 | 0.0348 | 0.0309 | 0.0494 | 0.0203 | 0.0262 | 0.0186 | 0.0314 | 0.0121 | 0.0162 |
| Bert4Rec | 0.0255 | 0.0399 | 0.0159 | 0.0206 | 0.0267 | 0.0453 | 0.0162 | 0.0221 | 0.0189 | 0.0325 | 0.0116 | 0.0159 |
| LightGCN | 0.0367 | 0.0577 | 0.0225 | 0.0293 | 0.0244 | 0.0421 | 0.0154 | 0.0211 | 0.0205 | 0.0355 | 0.0129 | 0.0177 |
| SASRec | 0.0337 | 0.0490 | 0.0213 | 0.0263 | 0.0342 | 0.0559 | 0.0216 | 0.0285 | 0.0190 | 0.0337 | 0.0118 | 0.0165 |
| BigRec | 0.0539 | 0.0774 | 0.0407 | 0.0493 | 0.0317 | 0.0522 | 0.0221 | 0.0299 | 0.0154 | 0.0169 | 0.0137 | 0.0142 |
| P5-TID | 0.0001 | 0.0001 | 0.0000 | 0.0000 | 0.0051 | 0.0076 | 0.0031 | 0.0039 | 0.0184 | 0.0263 | 0.0130 | 0.0155 |
| P5-SemID | 0.0689 | 0.0944 | 0.0442 | 0.0524 | 0.0374 | 0.0609 | 0.0231 | 0.0306 | 0.0202 | 0.0324 | 0.0131 | 0.0170 |
| P5-CID | 0.0678 | 0.0867 | 0.0544 | 0.0605 | 0.0349 | 0.0594 | 0.0225 | 0.0304 | 0.0219 | 0.0347 | 0.0140 | 0.0181 |
| LC-Rec | 0.0760 | 0.0940 | 0.0630 | 0.0690 | 0.0390 | 0.0590 | 0.0270 | 0.0330 | 0.0210 | 0.0320 | 0.0150 | 0.0180 |
| | (0.0009) | (0.0003) | (0.0005) | (0.0003) | (0.003) | (0.0048) | (0.0019) | (0.0024) | (0.0047) | (0.0086) | (0.0027) | (0.0038) |
| **STAR** (Our) | 0.0780 | 0.0960 | 0.0640 | 0.0700 | 0.0440 | 0.0670 | 0.0300 | 0.0380 | 0.0350 | 0.0450 | 0.0250 | 0.0280 |
| | (0.0008) | (0.0003) | (0.0011) | (0.0009) | (0.0011) | (0.0026) | (0.0003) | (0.0008) | (0.0020) | (0.0038) | (0.0015) | (0.0020) |
| Improvement | 2.44% | 2.51% | 1.11% | 1.25% | 13.83% | 14.41% | 12.58% | 13.05% | 65.12% | 40.42% | 73.57% | 58.23% |

Table 2: Overall performance comparison of search recommendation for both the status-quo competitive LLM-based baseline `LC-Rec` and our proposed **STAR**.

| Methodology | Arts | | | | Games | | | | Yelp | | | |
|---|---|---|---|---|---|---|---|---|---|---|---|---|
| | R@5 | R@10 | N@5 | N@10 | R@5 | R@10 | N@5 | N@10 | R@5 | R@10 | N@5 | N@10 |
| LC-Rec[4] | 0.00500 | 0.00900 | 0.00300 | 0.00400 | 0.07700 | 0.10500 | 0.05200 | 0.06100 | 0.0320 | 0.0340 | 0.0250 | 0.0260 |
| | (0.0015) | (0.0044) | (0.0009) | (0.0018) | (0.0086) | (0.0132) | (0.0067) | (0.0079) | (0.0048) | (0.0064) | (0.0038) | (0.0043) |
| **STAR** (Our) | 0.01400 | 0.02300 | 0.00800 | 0.01100 | 0.09900 | 0.13400 | 0.06600 | 0.07700 | 0.0390 | 0.0490 | 0.0300 | 0.0320 |
| | (0.0052) | (0.005) | (0.0038) | (0.0037) | (0.0058) | (0.0054) | (0.0038) | (0.0038) | (0.0107) | (0.015) | (0.0073) | (0.0085) |
| Improvement | 156.12% | 151.43% | 163.09% | 157.20% | 28.53% | 27.09% | 27.26% | 26.67% | 24.89% | 41.09% | 18.13% | 25.30% |

for 20K epochs, resulting in high codebook utilization and a low collision rate. During the token-embedding alignment stage and the subsequent supervised fine-tuning, we adapt `Qwen3-0.6B` via parameter-efficient fine-tuning (LoRA). Unless otherwise specified, all experiments are run on four NVIDIA H100 GPUs.

### 3.2.2 OVERALL PERFORMANCE ON RETRIEVAL TASKS.

Table 1 presents retrieval recommendation results across various datasets. The results demonstrate two key findings: (1) LLM-based generative recommenders consistently outperform traditional recommenders across all metrics, and (2) **STAR** achieves significant improvements over competitive baselines in both traditional recommender models and LLM-based generative models employing different identifier types. On the Arts dataset, **STAR** achieves notable performance gains over `LC-Rec`, with improvements ranging from 13.16% to 14.50% across recall and NDCG metrics. On the Games dataset, **STAR** demonstrates stronger improvements, outperforming `LC-Rec` by 20.33% to 21.69% across all evaluated metrics. On the Yelp dataset, **STAR** exhibits substantial superiority with improvements between 42.12% and 63.35%. These comprehensive results validate that **STAR** effectively addresses token embedding space misalignment by integrating structured linguistic semantics of semantic IDs into well-pretrained LLMs, thereby enhancing downstream recommendation performance.

### 3.2.3 OVERALL PERFORMANCE ON SEARCH TASKS.

Table 2 presents the search recommendation performance comparison between the competitive LLM-based baseline `LC-Rec` and our proposed **STAR** across three datasets. **STAR** demonstrates substantial improvements over `LC-Rec` across all metrics and datasets. On the Beauty dataset, **STAR** achieves remarkable performance gains, with improvements ranging from 331.94% to 361.90% across recall and NDCG metrics. Similarly, on the Instruments dataset, **STAR** outperforms `LC-Rec` by 57.81% to 102.56% across all evaluated metrics. On the Yelp dataset, **STAR** maintains consistent superiority with improvements between 23.67% and 60.23%. These comprehensive results validate

---

[4]The original `LC-Rec` implementation targeted only retrieval tasks. For search task adaptation, we combine the search query dataset with their auxiliary semantic alignment tasks and apply supervised fine-tuning to the base LLM following their established methodology.

that **STAR** effectively addresses limitations of existing LLM-based recommenders in search scenarios, demonstrating the robustness of our semantic ID approach across different recommendation domains and dataset characteristics. Additional experimental results for other datasets are provided in Appendix A.5.

### 3.3 ABLATION STUDY

**Two-stage full-prameter alignment vs. STAR alignment.** We additionally evaluate a full-model adaptation variant that updates all LM parameters on the semantic-alignment auxiliary tasks prior to finetuning on the search recommendation objective, as shown in table 7. This full-parameter ablation achieves performance comparable to our proposed **STAR** method (as shown in table 3), demonstrating *the primary performance gains of semantic alignment stem from injecting linguistic semantics into the new tokens rather than from broad backbone model adaptation*. The result validates our design rationale: targeted embedding updates achieve similar retrieval quality with substantially lower computational cost and reduced risk of overfitting associated with extensive backbone parameter modification.

Table 3: Two-stage alignment vs. **STAR** alignment

| Methodology | Beauty | | | |
|---|---|---|---|---|
| | R@5 | R@10 | N@5 | N@10 |
| Two-Stage Alignment | 0.03561 | **0.06323** | 0.01977 | **0.02865** |
| **STAR** (Our) | **0.03882** | 0.05282 | **0.02275** | 0.02735 |

**Data Scaling.** To test whether the gains from token–embedding alignment dissipate with more data, we vary the number of training samples from $10^3$ to $5 \times 10^3$. Across all four metrics (Recall@5/10, NDCG@5/10), STAR improves more rapidly than LC-Rec, yielding larger absolute and relative advantages at higher data volumes.

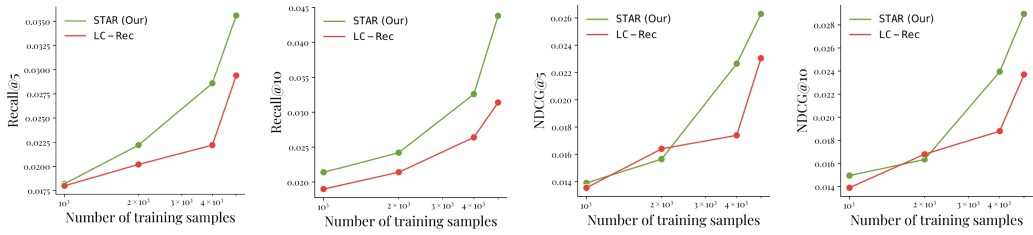

Figure 3: **Scaling with training set size.** Recall@5/10 and NDCG@5/10 versus the number of training samples for **STAR** (green) and LC-Rec (red). **STAR** consistently outperforms LC-Rec across data scales, with gaps that grow as the dataset enlarges, indicating that the benefits of token–embedding alignment neither vanish nor saturate in this regime.

**Candidate Pool Scaling.** To further assess the generalization capability of our proposed method beyond the standard top-K metrics, we conducted an additional evaluation in which we vary the candidate set size from 1 to 100 (unlike the primary benchmark—limited to Recall@5/10 and NDCG@5/10). As shown in Figure X, our method consistently surpasses LC-Rec across all candidate sizes, and the performance gap widens as the candidate pool expands. This trend indicates that STAR not only provides stronger top-K accuracy but also scales more robustly with task difficulty, demonstrating superior generalization and resilience compared with the baseline.

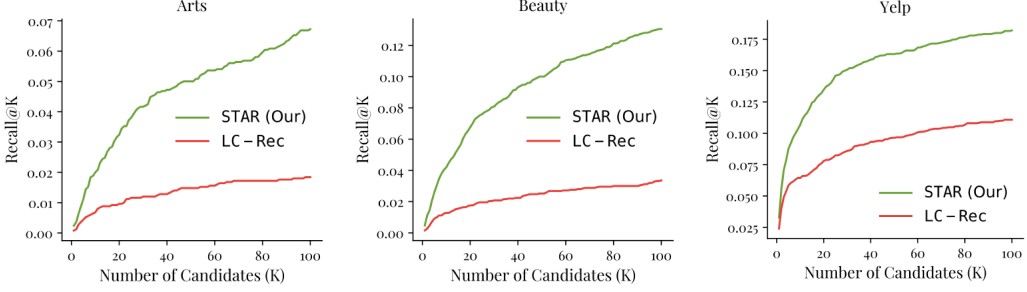

Figure 4: **Scaling with candidate pool.** Our method consistently surpasses LC-Rec across all candidate sizes, and the performance gap widens as the candidate pool expands.

## 3.4 Further Analysis

We analyze how initialization shapes the geometry of the Semantic-ID embedding subspace and how this geometry evolves under the same supervised fine-tuning stage. Taken together, the diagnosis in Figure 5 (initialization) and Figure 2-Right (post–fine-tuning spectrum) support our central claim: token-embedding alignment produces a *structured, linguistically grounded prior* that avoids collapse and remains non-degenerate after training. Figure 5 visualizes pairwise cosine similarities among well-pretrained vocabulary tokens and SemID tokens for three initialization schemes. In contrast to either random initialization or mean initialization strategy, our *token-embedding alignment* exhibits rich, differentiated structure within the SemID block together with coherent cross-block affinities to relevant lexical tokens. This pattern indicates that the aligned SemID vectors inherit *linguistic coordinates* from the pretrained space rather than merely occupying it, furnishing an informative starting point for downstream learning.

To explore whether a good prior persists after training, we stack the learned SemID token embeddings into $E_{\text{SemID}} \in \mathbb{R}^{|\mathcal{V}_{\text{SemID}}| \times d}$ and examine the singular-value spectrum $\sigma_i(E)$ (Figure 2-Right, for more results, check Appendix A.6). Starting from collaposed initialization leads to a rapidly decaying spectrum and a low effective rank, consistent with a near one-dimensioanl subspace that encodes little item-level diversity. By contrast, starting from our aligned prior yields a slower spectral decay and a markedly higher effective rank, signaling a non-degenerate SemID subspace with multiple active directions along which items differ. These results confirm that our method does more than avoid collapse at $t = 0$: it seeds directions that remain *useful* under downstream recommendation task supervised finetuning, enabling the model to carve a semantically meaning Semantic-ID token embedding subspace rather than re-learning from a degenerate start point.

## 4 Related Work

Recent work reframes recommendation as sequence generation, where models autoregressively decode item identifiers instead of relying on nearest-neighbor search in embedding space (Rege et al., 2023; Chen et al., 2025b). A key enabler is the use of vector-quantized autoencoders such as RQ-VAE (van den Oord et al., 2018; Zhang & Fu, 2025; Lee et al., 2022), which discretize items into Semantic IDs (SIDs) with hierarchical codebooks, providing compositional structure that allows language-model-style generation to capture fine-grained semantics of user histories. Building on this foundation, systems such as TIGER (Rajput et al., 2023) and LC-Rec (Zheng et al., 2024) demonstrate improved Recall/NDCG, while MTGR (Han et al., 2025), OneSearch (Chen et al., 2025a), and OneRec (Deng et al., 2025; Zhou et al., 2025) show scalable deployment in industry with cross-feature integration, keyword-enhanced quantization, and session-wise preference alignment. Beyond SID-based retrieval, LLM-driven knowledge-graph recommenders (Cai et al., 2025) further highlight the benefit of structured knowledge integration. We provide an extended discussion of related works in Appendix A.7.

## 5 Conclusion

We identify a fundamental token-embedding misalignment between newly introduced Semantic-ID tokens and pretrained language model vocabularies that significantly degrades generative recommendation systems. To address this, we propose a parameter-efficient pretraining approach **STAR** that selectively updates only the semantic-ID embeddings ($|V_{\text{semID}}| \times d$) while freezing the language model weights. This targeted alignment effectively aligns newly initialized Semantic ID token embeddings with well-pretrained LM's token embedding space. Extensive experiments demonstrate that **STAR** consistently outperforms mean-of-vocabulary initialization and auxiliary-task adaptation methods, yielding superior data efficiency and stronger top-$K$ retrieval for sequential recommendation and search across multiple datasets.

**Future Work.** While our study focuses on token embedding pretraining for language model adaptation, alternative approaches could encode semantic similarity directly within standard sequential models to potentially enhance performance. We believe that token-embedding alignment represents a promising approach for vocabulary expansion in domain adaptation, and we encourage future work to validate this method's effectiveness across diverse tasks and model architectures.

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

# A APPENDIX

## A.1 DATASETS

### A.1.1 RETRIEVAL DATASET

**Amazon Reviews 2023.** Our experiments employ the Amazon Reviews 2023 dataset (He & McAuley, 2016; Hou et al., 2024), which contains 571.54 million user reviews across 34 product categories spanning May 1996 to September 2023. Following prior work (He & McAuley, 2016; Zhang et al., 2019), we adopt the official 5-Core "pure-IDs" split to ensure sufficient interaction density (each user and item has $\geq 5$ interactions), improving both statistical reliability and reproducibility. In line with the sequential recommendation setting (Rajput et al., 2023), we further truncate user histories to a maximum length of 20, retaining only the most recent interactions. For sequential recommendation, we randomly sample four categories—`Arts`, `Beauty`, `Games`, and `Instruments`. We consider only text-based attributes (titles and descriptions) to simplify the setup.

**Yelp.** The Yelp Open Dataset (Yelp, 2025) is a subset of Yelp data that is intended for educational use. It provides real-world data related to businesses including reviews, photos, check-ins, and attributes like hours, parking availability, and ambience.

**Preprocessing.** Following prior work (He & McAuley, 2016; Zhang et al., 2019), we apply the standard 5-core filtering, removing users with fewer than five interactions and items with fewer than five associated users. In line with the sequential recommendation setting (Rajput et al., 2023), we truncate each user history to at most 20 events by keeping the most recent interactions. Summary statistics for all datasets are reported in Table 4.

**Evaluation Protocol and Metrics.** We adopt the standard leave-one-out protocol (Kang & McAuley, 2018b; Geng et al., 2023): for each user sequence, the last interaction is held out for testing, the penultimate interaction for validation, and the remainder for training. Recommendation quality is measured by Recall@$K$ and NDCG@$K$ with $K \in \{5, 10\}$.

### A.1.2 SEMI-SYNTHETIC SEARCH DATASET

Prior work (Ai et al., 2017; Shi et al., 2025) constructs "queries" through rule-based concatenation of Amazon category labels (removing stopwords and duplicates while excluding shallow categories), yielding taxonomy-like keyword strings (e.g., "photo digital camera lenses") that are coarse and fail to capture authentic user intents. To address this limitation, we employ large language models (LLMs) to generate nuanced, diverse queries that better reflect realistic search behavior. Specifically, we utilize the `GPT-OSS-20B` model as our base LLM and design a search query synthesis prompt (detailed in Appendix A.2.2). We generate five distinct queries per item, and manual inspection confirms that these LLM-generated queries exhibit superior quality compared to rule-based category strings. Due to computational resource constraints, we limit query dataset generation to five datasets: Arts, Beauty, Games, Instruments, and Yelp. For the Instruments dataset, we generate queries for all products, while for the remaining datasets, we generate queries for 5,000 randomly selected products[5]. We will publicly release these fine-grained synthetic query datasets to accelerate research in search and recommendation systems.

### A.1.3 DATASET STATISTICS

All dataset statistics are shown in table 4 and table 5.

---

[5]Within our data generation pipeline, we automatically remove malformed generations that do not follow the JSON format. Occasionally, the language model does not fully adhere to instructions and may generate fewer than the specified five queries. Given the rarity of this occurrence, we consider this acceptable and implement appropriate handling mechanisms.

Table 4: Summary Statistics for Retrieval Recommendation Datasets

| Dataset | # items | # User Interactions | Average Interaction Length |
|---|---|---|---|
| Arts | 20956 | 45141 | 8.658 |
| Beauty | 12101 | 22363 | 8.876 |
| Games | 16859 | 50546 | 8.962 |
| Instruments | 9922 | 24772 | 8.322 |
| Yelp | 20033 | 30431 | 10.396 |

Table 5: Summary Statistics for Search Recommendation Datasets

| Dataset | # items | # queries |
|---|---|---|
| Arts | 5000 | 24992 |
| Beauty | 5000 | 24991 |
| Games | 5000 | 24991 |
| Instruments | 9922 | 49564 |
| Yelp | 4999 | 24986 |

## A.2 PROMPT TEMPLATES

### A.2.1 PROMPT TEMPLATE: ITEM TITLE/DESCRIPTION ↔ SEMANTIC IDS

**Item Title/Description → Semantic IDs**[6] *(Title→SID)*

```
<system>
You are a helpful assistant.
<user>
Which item has the title: {{title}}?
<assistant>
{{ITEM SEMANTIC_ID}}
```

**Item Title/Description → Semantic IDs** *(Description→SID)*

```
<system>
You are a helpful assistant.
<user>
Can you tell me what item is described as {{description}}?
<assistant>
{{ITEM SEMANTIC_ID}}
```

**Item Title/Description → Semantic IDs** *(Title+Description→SID)*

```
<system>
You are a helpful assistant.
<user>
What item is called {{title}} and described as {{description}}?
<assistant>
{{ITEM SEMANTIC_ID}}
```

**Semantic IDs → Item Title/Description** *(SID→Title)*

```
<system>
You are a helpful assistant.
<user>
Could you please tell me what item {{ITEM SEMANTIC_ID}} is called?
<assistant>
```

```
{{title}}
```

**Semantic IDs → Item Title/Description** *(SID→Description)*

```
<system>
You are a helpful assistant.
<user>
Briefly describe item {{ITEM SEMANTIC_ID}}.
<assistant>
{{description}}
```

**Semantic IDs → Item Title/Description** *(SID→Title+Description)*

```
<system>
You are a helpful assistant.
<user>
What is the title and description of item {{ITEM SEMANTIC_ID}}?
<assistant>
{{title}}\n\n{{description}}
```

### A.2.2 PROMPT TEMPLATE: SEARCH QUERY TASK

**Synthetic Search Query Generation Prompt**

```
<system>
You are a helpful assistant.
<user>
You are an AI assistant specializing in generating realistic user
    search queries for products. Your task is to analyze product
    information and create diverse, natural-language search queries
    that potential customers might use when looking for a specific
    product.

Here's the JSON data for the product you need to analyze:

<product_json>
{{PRODUCT_JSON}}
</product_json>

Generate exactly 5 different user queries that are most likely to be
    used when searching for this specific product. Consider various
    aspects that users might search for, including:
- Brand
- Product type
- Specific features
- Use cases
- Common misspellings or alternative names

Ensure that:
- The queries use natural language and phrasing real users would
    likely employ
- The length and specificity of the queries vary
- Both broad and narrow search terms related to the product are
    included
- The queries are diverse and cover different aspects of the product
```

---

[6]Most of *Item Title/Description ↔ Semantic IDs* prompts and retrieval prompts are adapted from (Zheng et al., 2024).

```
Your output must be in pure JSON format, containing exactly 5 query
    objects. Do not include any explanations, analysis, or additional
    text. The output should follow this structure:

```json
[
{
\"query\": \"example search query 1\"
},
{
\"query\": \"example search query 2\"
},
{
\"query\": \"example search query 3\"
},
{
\"query\": \"example search query 4\"
},
{
\"query\": \"example search query 5\"
}
]
```
```

**Search Query Prompt (Template 1)**[7]

```
<system>
You are a helpful assistant.
<user>
As a recommender system, you are assisting a user who expresses a
    desire to obtain an item with the following characteristics: {
    query}. Please recommend an item that meets these criteria.
<assistant>
{{ITEM SEMANTIC_ID}}
```

**Search Query Prompt (Template 2)**

```
<system>
You are a helpful assistant.
<user>
The user wants a new item and searches for: {query}. Please select a
    suitable item that matches the search intent.
<assistant>
{{ITEM SEMANTIC_ID}}
```

**Search Query Prompt (Template 3)**

```
<system>
You are a helpful assistant.
<user>
Based on the user's current query {query}, please generate an item
    that matches the user's intent.
<assistant>
{{ITEM SEMANTIC_ID}}
```

---

[7]For brevity, we illustrate only three representative prompting templates.

### A.2.3 PROMPT TEMPLATE: RETRIEVAL TASK

---

**Retrieval Prompt (Template 1)**

```
<system>
You are a helpful assistant.
<user>
The user has interacted with items {{inters}} in chronological order.
    Can you predict the next possible item that the user may expect?
<assistant>
{{ITEM SEMANTIC_ID}}
```

---

**Retrieval Prompt (Template 2)**

```
<system>
You are a helpful assistant.
<user>
Based on the items that the user has interacted with: {{inters}}, can
    you determine what item would be recommended to the user next?
<assistant>
{{ITEM SEMANTIC_ID}}
```

---

**Retrieval Prompt (Template 3)**

```
<system>
You are a helpful assistant.
<user>
Here is the item interaction history of the user: {{inters}}, what to
    recommend to the user next?
<assistant>
{{ITEM SEMANTIC_ID}}
```

---

### A.3 BASELINE IMPLEMENTATION

For fair comparison, we adopt widely-used toolkits and official implementations to reproduce all baseline models.

**Traditional Recommenders.** We implement MF, HGN, Caser, BERT4Rec, LightGCN, and SASRec using the RecBole (Zhao et al., 2021) framework, which provides standardized implementations of classical recommendation models. Following prior work, we tune hyper-parameters based on the performance on a leave-one-out validation split. For each model, the configuration that yields the best validation performance is subsequently used for reporting test results. Since previous work compared these traditional baselines with LLM-based recommenders using large backbones (e.g., LLaMA-7B (Grattafiori et al., 2024)), their default model sizes are relatively large. For fair comparison with our main method (built upon the `Qwen3-Embedding-0.6B` backbone), we proportionally scale down the traditional recommender baselines by reducing their embedding dimensions and the number of layers (approximately one-tenth of the default size). Note that this scaling-down may weaken the absolute performance of these baselines compared to their default large-size configurations reported in prior work, but it provides a more equitable comparison in terms of computational budget.

**LLM-based Generative Recommenders.** For BIGRec and the P5 variants (P5-TID, P5-SemID, P5-CID), we rely on the official source code released by the original authors (Bao et al., 2023; Geng et al., 2023). We strictly follow their preprocessing pipelines and training procedures, while performing additional hyper-parameter search on the same leave-one-out validation split to ensure comparability. For a fair comparison with our main method (which is built upon the `Qwen3-Embedding-0.6B` backbone), we replace the original backbones with models of comparable scale: specifically, we adopt `LLaMA-3.2-1B` (Grattafiori et al., 2024) as the backbone for BIGRec and `T5-base` (Raffel et al., 2023) (0.2B) as the backbone for the P5 variants. For the P5 variants, we train for 20 epochs and select the best-performing checkpoint on the validation set for final evaluation.

## A.4 Our Proposed **STAR** Implementation

We utilize the pre-trained `Qwen3-Embedding-0.6B` encoder to extract semantic representations for items. The encoder processes item metadata including titles and descriptions to generate 1024-dimensional dense vectors that capture semantic similarities between items. We process text features of products by concatenating them as: [TITLE] [DESCRIPTION]. We set the maximum input sequence length as 2048. The final outputs are dense semantic embeddings: $z_i \in \mathbb{R}^{1024}$ for item $i$.

Our Residual Quantized Variational Autoencoder (RQ-VAE) follows the TIGER (Rajput et al., 2023) framework with carefully designed architectural specifications to ensure effective quantization of semantic representations. The encoder architecture consists of a 3-layer Multi-Layer Perceptron (MLP) with hidden dimensions of [1024, 512, 256], utilizing ReLU activation functions and applying a dropout rate of 0.1 between layers. The residual quantization mechanism employs four codebook layers, each containing 256 entries with 32-dimensional codes. This hierarchical quantization approach enables fine-grained representation of semantic information while maintaining discrete tokenization properties essential for language model integration. We trained the model for 20,000 epochs to achieve a high codebook utilization rate and minimize collision rates. To further prevent collisions where multiple items map to identical sequences of semantic IDs, we employed the Sinkhorn-Knopp trick used by LC-Rec (Zheng et al., 2024), which ensures uniform distribution of item semantics across codebook embeddings in the final layer.

The base language model employs `Qwen3-0.6B` with hidden dimension of 1024. The model architecture comprises 28 transformer layers supporting a maximum context length of 32,768 tokens. This configuration provides sufficient capacity for processing sequential recommendation tasks while maintaining computational efficiency. Parameter-efficient fine-tuning is implemented through Quantized Low-Rank Adaptation (QLoRA) with a rank of 8 and alpha value of 32. The LoRA adaptation applies a dropout rate of 0.05 and targets key projection matrices including q_proj, k_proj, v_proj, o_proj, gate_proj, up_proj, and down_proj. We also set LoRA modules to be saved as embed_tokens and lm_head, so that only the embedding layer and the language modeling head are preserved during training while other modules can remain frozen. This configuration enables efficient adaptation while preserving pre-trained knowledge.

We implement the token-embedding alignment stage of **STAR** by extending the Hugging Face TRL (HuggingFace, 2025) `SFTTrainer` to update only the Semantic-ID embedding matrix while freezing the LM backbone; the trainer consumes paired (title/description, SemID) examples and optimizes the embeddings as outlined in the pseudo code below. Unless otherwise stated, we train for 10 epochs with a learning rate of 1e-3 and a batch size 16.

## A.5 Additional Experiment Results

For completeness, we report extended experimental results that could not be included in the main text due to page limitation.

Table 6: Performance comparison on additional datasets (`Beauty` and `Instruments`) between the competitive LLM-based baseline `LC-Rec` and our proposed **STAR** method.

| Methodology | Beauty | | | | Instruments | | | |
|---|---|---|---|---|---|---|---|---|
| | R@5 | R@10 | N@5 | N@10 | R@5 | R@10 | N@5 | N@10 |
| LC-Rec | 0.00840 | 0.01160 | 0.00527 | 0.00629 | 0.01675 | 0.02362 | 0.01137 | 0.01356 |
| **STAR (Our)** | **0.03882** | **0.05282** | **0.02275** | **0.02735** | **0.03048** | **0.04784** | **0.01794** | **0.02349** |

---

**Algorithm 1:** Selective Token Embedding Training for STAR_SFT_TRAINER

---

**Input:** Pretrained model $\mathcal{M}$ with input embedding matrix $E \in \mathbb{R}^{V \times d}$; set of trainable token IDs
$\quad\quad \mathcal{T} \subseteq \{0, \ldots, V-1\}$
**Output:** Fine-tuned model $\mathcal{M}$ with updated embeddings for tokens in $\mathcal{T}$
**Initialization Phase:**
$\quad \mathcal{M}_{\text{backbone}} \leftarrow \text{FREEZE\_ALL\_PARAMETERS}(\mathcal{M} \setminus E)$ // Freeze backbone params
$\quad \text{INITIALIZE\_SELECTIVE\_EMBEDDING\_TRAINING}(\mathcal{M}, \mathcal{T})$
**Procedure** INITIALIZE\_SELECTIVE\_EMBEDDING\_TRAINING$(\mathcal{M}, \mathcal{T})$:
$\quad$ **Step 1:** Create binary mask $\mathbf{m} \in \{0, 1\}^V$ where:

$$m_i = \begin{cases} 1 & \text{if } i \in \mathcal{T} \\ 0 & \text{otherwise} \end{cases} \quad\quad\quad (3)$$

$\quad$ **Step 2:** Define selective gradient hook function:

$$\text{SELECTIVE\_GRADIENT\_HOOK}(\nabla E) = \nabla E \odot \mathbf{M} \quad\quad\quad (4)$$

$\quad$ where $\mathbf{M} \in \mathbb{R}^{V \times d}$ is $\mathbf{m}$ broadcasted to match $\nabla E$'s dimensions
$\quad$ **Step 3:** Register gradient hook on embedding matrix:
$\quad E.\text{REGISTER\_HOOK}(\text{SELECTIVE\_GRADIENT\_HOOK})$
**Training Phase:**
$\quad$ **for** *each training batch* $\mathcal{B}$ **do**
$\quad\quad \mathcal{L} \leftarrow \text{COMPUTE\_LOSS}(\mathcal{M}(\mathcal{B}))$                   // Forward pass
$\quad\quad \nabla E \leftarrow \text{BACKWARD\_PASS}(\mathcal{L})$                // Compute gradients
$\quad\quad \nabla E \leftarrow \text{SELECTIVE\_GRADIENT\_HOOK}(\nabla E)$   // Apply selective masking
$\quad\quad \text{UPDATE\_PARAMETERS}(E, \nabla E)$    // Update only selected embeddings

---

Table 7: Comparison across parameter efficiency, semantic alignment, and finetuning objective. Our TokEmb-Alignment method, **STAR**, aligns Semantic-ID tokens with the pretrained token-embedding space through lightweight training of embedding parameters only, maintaining the next-item recommendation objective without incorporating auxiliary optimization tasks.

| Method | Trainable parameters | Parameter Efficiency | Semantic Alignment | Finetuning Objective |
|---|---|---|---|---|
| LC-Rec | Full model | ✗ | ✓ | ✗ |
| Two-stage Alignment | Full model | ✗ | ✓ | ✓ |
| **STAR (ours)** | $|\mathcal{V}_{\text{SemID}}| \times D$ | ✓ | ✓ | ✓ |

## A.6 FURTHER ANALYSIS

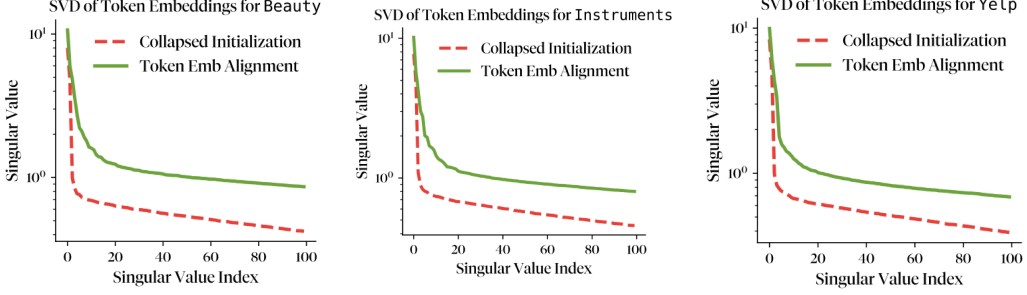

Figure 6: **Singular-value spectra of Semantic-ID embeddings.** Results for Beauty, Instruments, and Yelp datasets, showing consistent trends with the main paper (Figure 2-Right): token-embedding alignment produces slower spectral decay and higher effective rank than collapsed initialization.

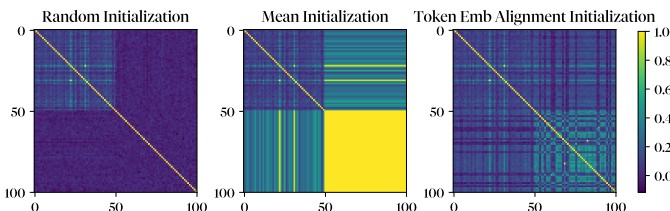

Figure 5: **Effect of initialization on Semantic-ID embeddings.** Pairwise cosine-similarity matrices of token embeddings for *Random Initialization*, *Mean-of-Vocabulary Initialization*, and our *Token-Embedding Alignment Initialization*. The upper-left block corresponds to 50 pretrained tokens and the bottom-right to 50 Semantic-ID tokens[9]. Random initialization (left) yields a noisy, unstructured SemID block with little affinity to the LM manifold—providing no linguistic prior and hindering adaptation. Mean-of-vocabulary places Semantic-ID vectors inside the LM manifold but collapses them into an almost uniform block (middle), making them semantically indistinguishable. Our alignment stage (right) yields a differentiated, linguistically grounded SemID subspace—an **informative, structured prior** for downstream supervised fine-tuning.

A.7   FULL RELATED WORK

**Generative Recommenders (GR).**   Generative retrieval reframes recommendation as sequence generation: rather than nearest-neighbor search in a shared embedding space (Rege et al., 2023; Chen et al., 2025b), a model autoregressively decodes item identifiers. TIGER realizes this by learning Semantic IDs (SIDs) and predicting the next SID from user history, improving Recall/NDCG (Rajput et al., 2023; Zheng et al., 2024). At scale, MTGR ships to production while preserving DLRM cross-feature signals (Han et al., 2025); OneSearch offers an end-to-end e-commerce system with keyword-enhanced quantization and preference-aware rewards (Chen et al., 2025a); and OneRec unifies retrieve-and-rank via session-wise generation and iterative preference alignment, with a companion report on large-scale deployment (Deng et al., 2025; Zhou et al., 2025). Beyond SID-based retrieval, LLM-driven knowledge-graph recommenders shows that structured knowledge integration can further enhance recommendation quality (Cai et al., 2025).

**RQ-VAE and Semantic IDs.**   Vector-quantized autoencoders (van den Oord et al., 2018; Zhang & Fu, 2025) have emerged as a central tool for learning discrete representations of items in generative recommender systems. In particular, Residual Quantized Variational Autoencoders (RQ-VAE) extend the original VQ-VAE framework by employing a hierarchy of residual codebooks to capture fine-grained semantic structure (Lee et al., 2022). Unlike conventional item IDs that treat items as independent symbols, Semantic IDs provide meaningful compositional structure, enabling autoregressive sequence models to predict future interactions more effectively. This combination of RQ-VAE-based discretization and language-model-style generation has become a foundation for state-of-the-art generative recommendation systems.

**Analogy to Dimensional Collapse in Contrastive Learning.**   The misalignment issue we identified is similar to the well-known phenomenon of *dimensional collapse* in contrastive learning Jing et al. (2021); Jiang et al. (2024). Unlike total collapse, where all embeddings converge to a single point, dimensional collapse restricts embeddings to a low-dimensional subspace as shown in Fig. 2, thereby eliminating fine-grained distinctions. Mean-of-vocabulary initialization exhibits this behavior: although Semantic-ID tokens may eventually spread apart during training, they start from a degenerate, low-rank configuration that lacks item-level diversity. This poor initialization substantially impedes learning efficiency and weakens downstream recommendation performance. Consistent with our findings, Jiang et al. (2024) also demonstrate that an appropriate initialization can significantly mitigate dimensional collapse, further underscoring the importance of embedding initialization in avoiding degenerate solutions.

---

[9]For better visualization, we randomly choose 50 tokens seperately from pretrained tokens or Semantic-ID tokens

## B  SUPPLEMENTARY MATERIALS

### ETHICS STATEMENT

This work adheres to the ICLR Code of Ethics.[10] Our research focuses on developing methods for token embedding alignment in generative recommenders. The experiments rely solely on publicly available datasets (Amazon Product Reviews and Yelp Open Dataset), which contain no personally identifiable information beyond what is publicly released. We do not foresee direct risks of harm to individuals or groups arising from this research. Nevertheless, as with all recommender systems, potential societal impacts include bias amplification and unintended reinforcement of popularity effects. We note these risks and emphasize that our contributions are methodological rather than application-specific; the proposed techniques can be combined with fairness-aware or debiasing mechanisms. No human subjects were involved, and no IRB approval was required.

### REPRODUCIBILITY STATEMENT

We are committed to facilitating reproducibility and transparency of our work. To this end, we intend to fully open source **all** of our code, data, and trained models after publication.

- **Code and Implementation:** We will provide an open-sourced codebase link to the full implementation and training scripts after publication.
- **Datasets:** All datasets used (Amazon Product Reviews and Yelp Open Dataset) are publicly available. Detailed preprocessing steps, including the 5-core filtering strategy and sequence truncation to length 20, are described in Appendix A.1. For semi-synthetic search query datasets we constructed, we will also publish it to accelerate research in search recommendation systems.
- **Model and Training Details:** Hyperparameters (learning rates, batch sizes, epochs, optimizer choices) and architectural specifications (Qwen3-0.6B configuration) are included in Section 3.1 and Appendix A.4.
- **Evaluation:** Metrics, evaluation protocols, and baselines are fully documented in Section 3.2.1 and Appendix A.3.

Together, these materials should enable independent researchers to reproduce our findings.

### THE USE OF LARGE LANGUAGE MODELS

Large language models were used in two ways during the preparation of this manuscript. First, we employed commercial LLMs solely to edit for clarity, grammar, and academic style, without altering the authors' intended meaning or contributions. Second, we used open-source LLMs—with a clearly specified prompting strategy A.2.2—to generate synthetic search recommendation datasets. In both cases, the authors exercised full oversight and accept responsibility for all claims, analyses, and conclusions.

---

[10] https://iclr.cc/public/CodeOfEthics

