# OpenReview forum: "STAR : Semantic-ID Token-Embedding Alignment For Generative Recommenders"
_ICLR.cc/2026/Conference — ICLR 2026 Conference Withdrawn Submission_

### Official Review · Reviewer_MFqC · 2025-10-26

**Soundness:** 3
**Presentation:** 3
**Contribution:** 2
**Rating:** 2
**Confidence:** 5

**Summary:**

Aiming at the token–embedding misalignment issue of semantic indices, this work proposes STAR, a lightweight alignment stage, to provide a proper initialization for the newly introduced semantic tokens. During the proposed alignment stage, only the embedding matrix corresponding to semantic tokens is optimizable, while the other components of the backbone language model are frozen. Extensive comparison with several traditional recommenders and generative recommender on both Amazon and Yelp datasets demonstrate the effectiveness of STAR.

**Strengths:**

1. The proposed method is well-introduced and easy to follow.
2. The proposed method can improve the sequential recommendation performance, compared to the leading baselines.

**Weaknesses:**

1. My main concern lies in the novelty of STAR, since there is no fundamental difference between STAR and LC-Rec (which is an evaluated baseline in this work). The claimed contributions, such as freezing the backbone LM, lie more in the perspective of engineering, rather than technological innovation.
2. The effectiveness of STAR is not sufficiently convincing. As introduced in this work, previous practices of semantic token initialization, including random embedding or mean embedding over vocabulary, are not appropriate for generative recommenders. Hence, from my perspective, STAR can be regarded as a better method to initialize embeddings of semantic tokens, which is able to combine with several different generative recommender models. By comparing the original baselines adopting random or mean initialization with the variant that adopts STAR initialization, the evaluation is more convincing when demonstrate the effectiveness.
3. The analyses of running time and memory are missing, which is important to support the 'lightweight' characteristics of STAR.
4. In the experiment, if LC-Rec and STAR are implemented with the same LM backbone? Moreover, an investigation on the influence of different LM backbones is recommended.
5. Organization of this manuscript is a little chaotic, of which Figure 1 and Figure 2 lack corresponding illustration in the main text.

**Questions:**

Please refer to the Weaknesses.

---

> ### Author Response · Authors · 2025-11-17
>
> We sincerely thank the reviewer for the constructive feedback. Below we address each concern in detail.
>
> ## Novelty of STAR vs. LC-Rec
> We would like to emphasize that our proposed STAR framework is fundamentally different from LC-Rec and offers broader applicability beyond recommendation, potentially serving as a general solution for domain adaptation scenarios that require integrating new vocabularies into pretrained language models. The core novelty of STAR lies in its two-stage learning paradigm. Rather than jointly optimizing vocabulary understanding and downstream objectives—as done in LC-Rec—STAR first learns semantically meaningful embeddings for the newly introduced tokens through auxiliary description tasks (e.g., item-description prediction), and then fine-tunes the model exclusively on the downstream tasks of interest. This decoupled optimization effectively mitigates interference between heterogeneous objectives and enables the model to better exploit pretrained linguistic knowledge.
>
> Through extensive experiments on five datasets spanning two distinct task types, STAR consistently outperforms LC-Rec, demonstrating that our design provides both conceptual and empirical advances rather than mere engineering adjustments.
>
> ## Effectiveness and Initialization Baselines
> To emphasize, random and mean initialization of token embeddings are widely adopted practices when extending the vocabulary of a pretrained language model. To the best of our knowledge, our work is the first to recognize that, for domain adaptation involving newly introduced vocabularies, a more effective strategy is to decouple the semantic alignment and downstream training stages. Instead of jointly optimizing over both the vocabulary description tasks and domain-specific objectives, STAR first leverages the description tasks to align new tokens within the pretrained model’s embedding space. The model is then fine-tuned exclusively on the downstream task. This two-stage process provides a principled and empirically effective alternative to random or mean initialization, yielding stronger generalization across various generative recommender architectures.
>
> To provide a comparison between different token embedding initialization techniques, we will add experiments evaluating random embedding initialization and integrate the results into the diagnostic analysis section soon.
>
> ## Backbone Consistency and Generalization Across LMs
> Yes, both LC-Rec and STAR are implemented using the same LM backbone (Qwen3-0.6B) to ensure a fair comparison. We agree that investigating the impact of different LM backbones is an important direction. Due to time and computational constraints, we did not include full parallel experiments with other backbones in the main paper. However, to address this concern, we additionally report the performance of our method on Llama3.2-1B  on the Yelp Dataset, which demonstrates consistent improvements and confirms the generalizability of our approach across different backbone architectures.
>
> The results show clear performance gains over
> | Method       | NDCG@5  | NDCG@10 | HR@5   | HR@10  |
> |--------------|---------|---------|--------|--------|
> | LC-Rec       | 0.02040 | 0.03340 | 0.01330 | 0.01750 |
> | STAR (Ours)  | **0.03790** | **0.05150** | **0.02670** | **0.03100** |
>
> These results confirm that our approach generalizes effectively across model backbones.
>
> ## Manuscript Organization and Figures
> We have refined the organization of the manuscript. Figure 1 and Figure 2 are now clearly referenced and discussed in the main text to improve readability and conceptual flow.

---

> ### Author Response · Authors · 2025-11-19
>
> To respond to the concern of the running time efficiency and the novelty of our proposed method, we add new experiments to compare different token embedding initialization methods in the following:
>
> | Yelp Retrieval Task | R@5 | R@10 | N@5 | N@10 |
> |---------------------|--------------|--------------|--------------|--------------|
> | Mean Initialization | 0.0140 | 0.0206 | 0.0098 | 0.0120 |
> | Random Initialization | *0.0302* | *0.0342* | *0.0225* | *0.0238* |
> | Vanilla Data Mixture (LC-Rec) | 0.0215±0.0047 | 0.0317±0.0086 | 0.0146±0.0027 | 0.0179±0.0038 |
> | STAR Initialization | **0.0355±0.0020** | **0.0446±0.0038** | **0.0254±0.0015** | **0.0283±0.0020** |
>
> The results demonstrate the effectiveness of our proposed STAR token-embedding initialization method. Our STAR method provide consistent performance boost compared with random initialization method and also with data-mixture method (LC-Rec).
>
> We also report the training time for each method in the following table:
>
> | **Method**                       | **Training Time** (RTX 4090 * 2) | **Extra Cost**                       |
> | -------------------------------- | ----------------- | ------------------------------------ |
> | Random Inititialization                      | 14h 28m 11s       | —                                    |
> | Mean Initialization                        | 14h 40m 26s       | —                                    |
> | **STAR Initialization (Ours)** | **15h 11m 36s**   | **~30 min** Token-Embedding Alignment |
> | **LC-Rec (Mixture Training)**  | **16h 41m 50s**   | **~2h** supervised fine-tuning       |
>
>
> As shown in our efficiency analysis, our method avoids supervised fine-tuning on the full set of semantic-meaning auxiliary tasks. Instead, it only performs lightweight alignment on the newly introduced token embeddings with respect to the pretrained LLM vocabulary. This design reduces the additional training cost to merely **30 minutes**. In contrast, LC-Rec’s data-mixture strategy requires an **extra 2 hours** of fine-tuning over all auxiliary semantic-meaning tasks. Overall, our approach achieves a **4x improvement** in time efficiency while simultaneously delivering substantially stronger performance ( **65.12% gain** in Recall@5 compared with LC-Rec).

---

> ### Comment · Reviewer_MFqC · 2025-11-22
>
> Dear authors, first of all, thanks for your reply to my questions and additional results. I'm willing to raise my score to 4, while my main concern, i.e., the novelty of STAR, remains outstanding. Hence, I think 4 is a reasonable rating for the current version of this work. Best wishes and good luck.

---

> > ### Author Response · Authors · 2025-11-26
> >
> > Thank you very much for your follow-up and for raising your score. We sincerely appreciate your time and effort, and also your constructive feedback, which can help us improve the work. We understand that concerns about the novelty of STAR remain unclear from your end.
> > ### We aim to address all your concerns and clarify your misunderstanding about our novelty, and provide a comparison with LC-Rec.
> > 1. The research focus is different: a) LC-Rec is a vanilla data mixture method for Adapting Large Language Models by Integrating Collaborative Semantics for Recommendation. b) Our STAR dives deeper and aims to solve the catastrophic manifold distortion due to the inconsistency between the inappropriate new token initialization and the well-pretrained existing token embedding manifold.
> > 2. Proposed Approach is different a) LC-Rec proposed a LLM-based method for recommendation, and utilized a VQ codebook allows similarity and avoid ID conflicts in index allocation b) We proposed a pipeline with a novel token embedding manifold adaptive insertion algorithm that solves the catastrohpic manifold distortion We understand this misunderstanding may comes from we adapt the auxiliary task design of LM-Rec to research our problem and propose our novel methods based on the previous work. Although LM-Rec is a great work that illuminates lots of works in this area, the only similarity in novelty we can think of is that we use their auxiliary task.
> >
> > In our common response, we add a clearer explanation to emphasize the conceptual novelty of our method, and we will also refine the manuscript to clearly articulate the distinctions, motivations, and also much broader applicability of STAR in the camera-ready version (if accepted). Thank you again for your consideration and for helping us improve the work.

---

### Official Review · Reviewer_45ci · 2025-10-30

**Soundness:** 2
**Presentation:** 3
**Contribution:** 2
**Rating:** 4
**Confidence:** 3

**Summary:**

This paper addresses the token-embedding misalignment problem in generative recommender systems that adapt pre-trained language models for sequential recommendation. The authors identify that when Semantic-ID tokens (discrete identifiers produced by RQ-VAE) are integrated into a pre-trained language model's vocabulary, the common mean-of-vocabulary initialization collapses all new tokens into an undifferentiated region in the embedding space, stripping item-level semantics. To remedy this, the authors propose STAR, a lightweight alignment stage that freezes the language model backbone and updates only the Semantic-ID token embeddings through paired supervision (item descriptions → Semantic-ID sequences). Experiments on multiple datasets show consistent improvements in top-K retrieval metrics over strong baselines.

**Strengths:**

S1. Well-Motivated Problem with Clear Presentation
The paper identifies a practical issue in adapting language models for generative recommendation and presents it clearly with effective visualizations, making the problem accessible and the solution intuitive.

S2. Simple and Parameter-Efficient Method
STAR updates only $|V_{\text{SemID}}| \times D$ parameters ($\sim$0.8M, representing $\sim$0.13\% of full model), making it computationally efficient and easy to integrate into existing generative recommendation pipelines without architectural modifications.

S3. Comprehensive Experimental Validation
Experiments cover 9 datasets across multiple domains (Amazon, Yelp), both retrieval and search tasks, with consistent improvements over strong baselines (13-63\% gains). Ablation studies on data scaling and alternative designs strengthen the empirical contribution.

**Weaknesses:**

W1. Weak Theoretical Foundation and Potentially Misdiagnosed Problem.
The paper lacks theoretical justification for why "token-embedding misalignment" is the fundamental problem, rather than simply insufficient content information integration.

W2. Unfair Baseline Comparison and Missing Critical Ablations
STAR and LC-Rec are fundamentally similar—both use item descriptions to train Semantic-ID embeddings, differing mainly in training schedule (pre-train vs. joint) and parameter updates (frozen vs. full). The paper criticizes LC-Rec for "memorizing all items" but STAR does the same in its alignment stage.

W3. Unaddressed Scalability and Practical Deployment Issues
The paper introduces an extra training stage but reports no wall-clock time, convergence analysis, or FLOPs comparison. Critical practical concerns are ignored: (1) What if alignment and downstream stages use different LMs (e.g., Llama vs. Qwen)? (2) How to handle new items in dynamic catalogs without full retraining? (3) Cross-model generalization (e.g., align with 0.6B, deploy with 7B)? Experiments only use small models (0.6B) and datasets (max 5K samples), limiting generalizability to industrial-scale systems with millions of items and interactions.

**Questions:**

See weaknesses

---

> ### Author Response · Authors · 2025-11-17
>
> We thank the reviewer for the thoughtful and constructive feedback. Below we address each concern in detail.
>
> ## On Baseline Fairness and Differences Between STAR and LC-Rec
>
> We would like to clarify that our proposed method is fundamentally different from LC-Rec in both formulation and optimization. LC-Rec augments the LLM’s token vocabulary with semantic IDs without any fine-grained control on the novel token embedding initialization, but just mixes the optimization goals given the auxiliary (item-description) and recommendation objectives to let the model indirectly learn semantic information of these semantic IDs during fine-tuning.
>
> In contrast, our approach explicitly formulates the alignment as a token-embedding initialization problem. We first leverage the auxiliary item-description tasks solely to obtain semantically meaningful token embeddings, and then fine-tune the LLM exclusively on any downstream recommendation objective (retrieval or search tasks). This decoupled two-stage training paradigm avoids optimization interference between auxiliary and main objectives, allowing the model to fully utilize the semantically enriched embeddings for downstream learning.
>
> Empirically, this design leads to consistent state-of-the-art performance across five benchmarks for both retrieval and search tasks, demonstrating superior generalization and stability compared to LC-Rec.
>
>
> ## On Scalability, Deployment Considerations, and Model Variants
>
> **What if alignment and downstream stages use different LMs (e.g., Llama vs. Qwen)?**
>
> For the first concern, the alignment stage is inherently tied to the backbone LLM’s tokenizer and token-embedding space. Since the aligned Semantic-ID embeddings are learned within a specific model’s vocabulary (e.g., Qwen3-0.6B), they cannot be directly transferred to another model series (e.g., LLaMA), which uses a distinct tokenizer and embedding basis.
>
> **How to handle new items in dynamic catalogs without full retraining?**
>
> For the second concern, handling dynamic catalogs with new items is indeed an important direction for future work. Our current focus is on studying the semantic alignment and initialization mechanisms, and extending them to incremental or continual learning scenarios would be an exciting next step for the community.
>
> **Cross-model generalization (e.g., align with 0.6B, deploy with 7B)?**
>
> For the third concern, we have not yet explored cross-model generalization (e.g., aligning on a 0.6B model and deploying on a 7B model) to ensure a controlled and fair comparison with LC-Rec and other baselines. We agree that this is a promising future direction to evaluate the transferability of the learned Semantic-ID embeddings across model scales.
>
> Finally, while our experiments are conducted on small language models (Qwen3-0.6B) due to computational constraints, we evaluated across multiple datasets and task types—from Amazon and Yelp platforms, covering both retrieval and search tasks. The dataset size ranges from 5K to 50K user interactions and up to 21K items, demonstrating the robustness and generality of our findings across diverse data regimes.

---

> ### Author Response · Authors · 2025-11-26
> **Dear Reviewer 45ci, Have Your Concerns Been Addressed?**
>
> Dear Reviewer 45ci,
>
> Thank you very much for the time and care you invested in reviewing our submission, as well as for the insightful suggestions you provided. We have incorporated the key new experiments and added detailed explanations throughout the paper to address your concerns.
>
> Could you please let us know if there are any remaining issues or additional improvements you would recommend? Your feedback is invaluable to us as we work to strengthen our work.
>
> Sincerely, The Authors

---

### Official Review · Reviewer_zpXt · 2025-11-01

**Soundness:** 1
**Presentation:** 2
**Contribution:** 2
**Rating:** 2
**Confidence:** 4

**Summary:**

This paper studies the initialization of semantic ID embeddings in LLM-based generative recommendation. The authors identify a key issue in existing approaches: embedding collapse in newly added semantic tokens, which prevents the recommender from fully utilizing the semantic priors of LLMs and reduces item-level distinctions. To address this, they propose a lightweight alignment method that learns token embeddings grounded in the existing vocabulary. Experiments on five real-world datasets show that the proposed method produces more informative and linguistically meaningful embeddings, improving both search and recommendation performance.

**Strengths:**

1. The paper focuses on semantic ID embedding learning, a fundamental problem in generative recommendation. The motivation is clear and relevant.

2. The authors identify a concrete limitation of existing initialization methods, where semantic token embeddings collapse and lead to suboptimal results.

3. The proposed lightweight alignment method is well designed and validated through experiments on five real-world datasets.

**Weaknesses:**

1. The comparison between standard sequential methods and language adaptation methods is not fully convincing. Although sequential models require more data and lack explicit semantics, they typically use smaller transformers (such as TIGER) and train much faster. Stronger theoretical or empirical evidence is needed to support the claimed advantages.

2. The preliminaries and diagnostic analyses overlook random embedding initialization, which could also mitigate embedding collapse.

3. The paper lacks sufficient discussion or empirical comparison with textual ID–based generative recommendation methods, such as IDGenRec [1].

4. The claims about generalization ability and sample efficiency are not well supported by the experiments. In addition, Figure 4 only shows distinctions among newly introduced token embeddings, which makes it difficult to verify that the aligned embeddings are linguistically grounded as claimed.

[1] Tan et al., Idgenrec: Llm-recsys alignment with textual id learning. SIGIR'24

**Questions:**

1. Do all LLM-based generative recommenders in the experiments share the same backbone (e.g., Qwen3-0.6B)? Consistent backbones are necessary for a fair comparison between random or mean initialization and the proposed STAR.

2. Is there any comparison of data efficiency between STAR and standard sequential models to substantiate the claimed improvement?

---

> ### Author Response · Authors · 2025-11-17
>
> We thank the reviewer for the constructive feedback and address each concern point-by-point below.
>
> ## Comparison Between Sequential Models and Language Adaptation Methods
>
> We appreciate the reviewer’s thoughtful observation regarding the computational efficiency of standard sequential recommenders such as TIGER. Indeed, sequential models trained from scratch typically employ smaller transformer architectures. However, these models lack explicit semantic grounding, which fundamentally limits their ability to generalize beyond collaborative-filtering signals—particularly in settings that require natural language understanding, such as **search or language-based retrieval**.
>
> In contrast, language-adaptation methods are able to leverage the pretrained linguistic priors of large language models, yielding richer semantic representations and stronger cross-task generalization. This explains why LC-Rec establishes a strong baseline compared with purely sequential approaches. As demonstrated in our main paper, our proposed method further builds on this advantage and achieves state-of-the-art results on **both sequential recommendation benchmarks (Table I) and search-oriented retrieval tasks (Table II)**, highlighting the benefits of grounding semantic IDs within a pretrained language-model backbone.
>
>
> ## Discussion of Random Embedding Initialization
> We agree that the comparsion with random embedding initialization is necessary. We will add experiments evaluating random embedding initialization and integrate the results into the diagnostic analysis section.
>
>
> ## Comparison with IDGenRec
> We thank the reviewer for pointing us to textual ID–based generative recommendation methods such as IDGenRec [1]. Although our work focuses on semantic-ID initialization rather than textual ID generation, these approaches are complementary. We will cite this line of work and include a discussion in the revised manuscript.
>
> ## Evidence Supporting Generalization Ability
> To further assess the generalization ability of our method relative to LC-Rec, we conducted an additional evaluation specifically aimed at examining generalization from the perspective of metric scale. Instead of relying solely on the standard metrics (Recall@5, Recall@10, NDCG@5, NDCG@10), we varied the number of candidates from 1 to 100 to understand how performance behaves under different metric scales. As shown in Figure 4 of the revised manuscript, our method consistently outperforms LC-Rec across all candidate sizes, and the performance gap widens as the candidate pool increases, providing further evidence of the superior generalization capability of our approach.
>
> We also construct experiments with different LM backbones(Llama3.2-1B) on the Yelp Dataset. The results show clear performance gains over
> | Method       | NDCG@5  | NDCG@10 | HR@5   | HR@10  |
> |--------------|---------|---------|--------|--------|
> | LC-Rec       | 0.02040 | 0.03340 | 0.01330 | 0.01750 |
> | STAR (Ours)  | **0.03790** | **0.05150** | **0.02670** | **0.03100** |
>
> These results confirm that our approach generalizes effectively across model backbones and different metric range.
>
>
> ## Q1: Whether All LLM-Based Generative Recommenders Share the Same Backbone
>
> For **traditional item tokenization methods** (BigRec, P5-TID, P5-SemID, P5-CID), we intentionally preserve each method’s **original backbone**. This follows established practice in LLM-based recommendation, where the backbone is an integral component of the method’s modeling formulation—for example, P5’s encoder–decoder joint encoder and BigRec’s instruction-tuned backbone used to construct oracle embeddings.
> Replacing these architectures with the decoder-only Qwen3-0.6B would substantially change their modeling behavior and therefore **would not constitute a faithful or fair reproduction** of the original methods.
> At the same time, to ensure cross-paradigm fairness, we select baselines whose backbone **parameter scales are comparable to Qwen3-0.6B**. This allows us to focus on the central question of interest—the difference between **traditional item tokenization and SemanticID**—without confounding the comparison with model size disparities.
>
> For **SemanticID-based methods** (LC-Rec, STAR), both are implemented **using the same backbone** (Qwen3-0.6B), ensuring a fully controlled and fair comparison when studying (1) the effect of SemanticID (2) the role of alignment and (3) different initialization strategies.
>
> ## Q2: Data Efficiency Comparison with Standard Sequential Models
> We agree that a thorough comparison against sequential models like TIGER would further support claims about data efficiency. We will add TIGER comparison experiments in analysis part soon. We note, however, that the backbone architectures differ significantly (LLM vs. small transformer), and such comparisons must be interpreted carefully to avoid misleading conclusions.

---

> ### Author Response · Authors · 2025-11-19
>
> To respond to the concern about the comparison with token embedding random initialization, we add new experiments to compare different token embedding initialization methods in the following:
>
> | Yelp Retrieval Task | R@5 | R@10 | N@5 | N@10 |
> |---------------------|--------------|--------------|--------------|--------------|
> | Mean Initialization | 0.0140 | 0.0206 | 0.0098 | 0.0120 |
> | Random Initialization | *0.0302* | *0.0342* | *0.0225* | *0.0238* |
> | Vanilla Data Mixture (LC-Rec) | 0.0215±0.0047 | 0.0317±0.0086 | 0.0146±0.0027 | 0.0179±0.0038 |
> | STAR Initialization | **0.0355±0.0020** | **0.0446±0.0038** | **0.0254±0.0015** | **0.0283±0.0020** |
>
> The results demonstrate the effectiveness of our proposed STAR token-embedding initialization method. Our STAR method provide consistent performance boost compared with random initialization method and also with data-mixture method (LC-Rec).
>
> We also report the training time for each method in the following table:
>
> | **Method**                       | **Training Time** (RTX 4090 * 2) | **Extra Cost**                       |
> | -------------------------------- | ----------------- | ------------------------------------ |
> | Random Inititialization                      | 14h 28m 11s       | —                                    |
> | Mean Initialization                        | 14h 40m 26s       | —                                    |
> | **STAR Initialization (Ours)** | **15h 11m 36s**   | **~30 min** Token-Embedding Alignment |
> | **LC-Rec (Mixture Training)**  | **16h 41m 50s**   | **~2h** supervised fine-tuning       |
>
>
> As shown in our efficiency analysis, our method avoids supervised fine-tuning on the full set of semantic-meaning auxiliary tasks. Instead, it only performs lightweight alignment on the newly introduced token embeddings with respect to the pretrained LLM vocabulary. This design reduces the additional training cost to merely **30 minutes**. In contrast, LC-Rec’s data-mixture strategy requires an **extra 2 hours** of fine-tuning over all auxiliary semantic-meaning tasks. Overall, our approach achieves a **4x improvement** in time efficiency while simultaneously delivering substantially stronger performance ( **65.12% gain** in Recall@5 compared with LC-Rec).

---

> ### Author Response · Authors · 2025-11-26
> **Dear Reviewer zpXt, Have Your Concerns Been Addressed?**
>
> Dear Reviewer zpXt,
>
> Thank you very much for the time and care you invested in reviewing our submission, as well as for the insightful suggestions you provided. We have incorporated the key new experiments and added detailed explanations throughout the paper to address your concerns.
>
> Could you please let us know if there are any remaining issues or additional improvements you would recommend? Your feedback is invaluable to us as we work to strengthen our work.
>
> Sincerely, The Authors

---

### Official Review · Reviewer_h4rs · 2025-11-04

**Soundness:** 3
**Presentation:** 4
**Contribution:** 3
**Rating:** 4
**Confidence:** 4

**Summary:**

This paper introduces a simple yet effective approach for aligning semantic-ID token embeddings with the token space of large language models. By using the learned token embeddings to initialize semantic tokens, these tokens can be readily adopted for downstream tasks. Through comprehensive evaluation, the authors demonstrate the effectiveness of their method.

**Strengths:**

- The paper is well-written and well-organized.
- It poses and discusses a foundational question regarding the misalignment issue in token embedding spaces.
- The experiments are well-designed.

**Weaknesses:**

- This alignment/initialization may not be necessary for typical industrial generative recommenders, such as TIGER and OneRec, where both parameters and token embeddings are trained from scratch and no language tokens are included in the vocabulary.
- There are some typos. For example, in Line 289, "... we randomly pick five categories ...", but only four are listed. Additionally, in Table 1 (Line 334), "P5-SimID" should be "P5-SemID".
- The proposed alignment method appears to be similar to the mutual prediction alignment introduced in LC-Rec. The only apparent difference seems to be whether the model parameters are fixed or not.
- Although the proposed alignment strategy could be model-agnostic, the experiments are conducted solely on a single language model, Qwen3-0.6B. This limits the generalizability of the method.

**Questions:**

- There is a contradictory observation between this paper and the baseline LC-Rec. On one hand, this paper argues that "the primary performance gains of semantic alignment stem from injecting linguistic semantics into the new tokens rather than from broad backbone model adaptation." On the other hand, LC-Rec (Table IV) shows in its ablation study that all alignment tasks contribute to performance improvement. Why do alignment tasks aimed at broad backbone model adaptation provide no contribution or even a negative impact on model performance in your experiments?
- Why is LC-Rec inferior to P5-SemID and P5-CID in most cases in Table 1? Considering that LC-Rec uses SemanticID and applies alignment, does this observation suggest that the semantic alignment is less effective?
- What is the maximum item sequence length used for the sequential recommendation comparison?

---

> ### Author Response · Authors · 2025-11-17
>
> We sincerely thank the reviewer for the constructive feedback. Below, we provide point-by-point responses and clarifications.
> ## Necessity of alignment/initialization for industrial generative recommenders (e.g., TIGER, OneRec)
>
> Our motivation for incorporating recommendation capabilities into a pretrained language model—rather than training a small sequential model such as TIGER or OneRec from scratch—is twofold:
> 1. Leveraging natural language priors:
>  As demonstrated in LC-Rec, LLMs provide rich semantic priors that enable more informative representations of semantic IDs. Models such as TIGER and OneRec, trained purely on collaborative signals, cannot capture such semantics.
>
>
> 2. Unifying retrieval and search:
>  Sequential recommenders do not generalize to natural-language search scenarios, which require linguistic understanding. Our modeling framework enables a unified treatment of retrieval and search, and our token-embedding alignment yields consistent performance gains across both tasks.
>
>
> Finally, the adoption of language-model–based systems is an accelerating trend in industry (e.g., OneSearch, Cai et al., 2025). Thus, semantic token initialization remains relevant and increasingly important.
> ## Refine in category listing and Table 1
> We thank the reviewer for identifying these issues. We have corrected the category count on Line 289 and fixed “P5-SimID” → “P5-SemID” in Table 1.
> ## Similarity to LC-Rec’s mutual prediction alignment
>
> We would like to emphasize the simplicity and effectiveness of our proposed alignment method, as also noted by reviewer MFqC. Our token-embedding alignment can be interpreted as a semantic initialization strategy for incorporating novel token embeddings into a well-trained LLM. Unlike LC-Rec, which jointly fine-tunes on both auxiliary and recommendation tasks (misleads the optimization focus), our approach leverages the auxiliary tasks solely to obtain semantically meaningful token embeddings before fine-tuning exclusively on the recommendation tasks. As shown in Table I (retrieval) and Table II (search), this design achieves state-of-the-art performance across both settings.
>
> ## Generalizability beyond Qwen3-0.6B
>
> We conducted additional experiments on Yelp using a different LLM backbone (Llama3.2-1B). The results show clear performance gains over
> | Method       | NDCG@5  | NDCG@10 | HR@5   | HR@10  |
> |--------------|---------|---------|--------|--------|
> | LC-Rec       | 0.02040 | 0.03340 | 0.01330 | 0.01750 |
> | STAR (Ours)  | **0.03790** | **0.05150** | **0.02670** | **0.03100** |
>
> These results confirm that our approach generalizes effectively across model backbones.
>
> ## Observation between this paper and the baseline LC-Rec
> We would like to clarify that the two-stage alignment setting in Table IV is conceptually different from the LC-Rec setup in Tables I and II. Specifically, LC-Rec jointly optimizes both semantic-alignment auxiliary tasks and recommendation tasks in a mixed training scheme, whereas our approach leverages the auxiliary tasks only for token-embedding alignment, followed by fine-tuning solely on recommendation tasks. In other words, our method can be viewed as providing a stronger token-embedding initialization mechanism, as also noted by Reviewer MFqC.
>
> The goal of Table IV is to show that this lightweight token-embedding initialization achieves comparable or even superior performance to FULL-parameter fine-tuning pipelines that first train on auxiliary tasks and then fine-tune on recommendation objectives. Furthermore, while LC-Rec’s dataset-mixing approach introduces linguistic knowledge into the model, it inevitably dilutes the optimization focus on the core recommendation objective, which explains why broad backbone adaptation in LC-Rec contributes little—or even negatively—to final performance in our experiments.
>
> ## Why is LC-Rec inferior to P5-SemID and P5-CID in most cases in Table 1?
>
> We would like to clarify that the assumption that LC-Rec is inferior to P5-SemID or P5-CID is inaccurate.
> Across all three datasets (Arts, Games, Yelp) and all evaluation metrics, SemanticID-based methods (LC-Rec and STAR) consistently outperform the P5 variants:
> + LC-Rec > P5-SemID in 8 out of 12 metric–dataset combinations
> + LC-Rec > P5-CID in 9 out of 12 combinations
> + STAR achieves the best performance on all metrics
>
> This trend directly demonstrates that SemanticID is substantially more effective than the traditional P5 item tokenization scheme. Therefore, the observed differences in Table 1 do not indicate that semantic alignment is ineffective; rather, they strongly support the effectiveness of SemanticID as a superior item representation for LLM-based recommendation.
>
> ## Maximum item sequence length in sequential recommendation
> As detailed in Appendix A.1, we follow standard practice: each user’s interaction history is truncated to the 20 most recent items.

---

> ### Author Response · Authors · 2025-11-26
> **Dear Reviewer h4rs, Have Your Concerns Been Addressed?**
>
> Dear Reviewer h4rs,
>
> Thank you very much for the time and care you invested in reviewing our submission, as well as for the insightful suggestions you provided. We have incorporated the key new experiments and added detailed explanations throughout the paper to address your concerns.
>
> Could you please let us know if there are any remaining issues or additional improvements you would recommend? Your feedback is invaluable to us as we work to strengthen our work.
>
> Sincerely,
> The Authors

---

### Author Response · Authors · 2025-11-17

We sincerely thank all reviewers for their thoughtful feedback and constructive comments. We greatly appreciate the time and effort you devoted to carefully evaluating our work. Your insights have been invaluable in helping us improve the quality and clarity of our paper.
We would like to emphasize the core innovations of our work, which were also recognized as key strengths by all reviewers.
- **Well-Motivated Problem Definition (45ci, zpXt)**: The paper identifies and articulates the token-embedding misalignment issue in LLM-based generative recommenders and discusses it with clarity and strong motivation.
- **Simple and Effective Method (zpXt, 45ci, MFqC)**: The proposed STAR alignment method is simple, lightweight, and parameter-efficient, updating only a small portion of the model while delivering strong performance improvements.
- **Foundational Insight into Embedding Misalignment (h4rs, zpXt, MFqC)**: The work raises and analyzes a core question regarding the token-embedding-space misalignment problem, highlighting a meaningful limitation of existing initialization approaches.
- **Comprehensive Experimental Validation (h4rs, 45ci, zpXt, MFqC)**: Experiments are extensive and well-designed, covering multiple datasets and both retrieval and search tasks, consistently demonstrating the effectiveness of the proposed method.
- **Empirical Performance Gains (45ci, MFqC)**: The proposed method shows clear improvements in top-K retrieval and search performance over strong baselines.
- **Clear and Well-Written Presentation (h4rs, 45ci, MFqC)**: The paper is well-written, well-organized, and easy to follow, presenting the motivation, method, and results clearly.

---

### Author Response · Authors · 2025-11-26

# Novelty Emphasis

Sincerely thank you for the time and effort you have devoted to reviewing our work. We would like to highlight the significance and key contributions of our proposed methodology.

We proposes a novel token embedding manifold adaptive insertion algorithm, which aims for token-embedding expansion under manifold-preserving constraints. Our method allows online vocabulary expansion on any well-pretrained LLM without catastrohpic manifold distortion due to the inconsistence between the inappropriate new token initialization and the well-pretrained existing token embedding manifold.

Existing models severely struggle when expand an LLM's vocabulary: either a randomly initialized embedding or a mean initialized embedding for a new token tends to distort the entire vocabulary manifold: Random initialization introduces large manifold-distance deviations that propagate through training, mean initialization collapses the new token into an overly generic representation, distorting neighborhood structure and degrading downstream performance.

We address this by explicitly aligning the new token embeddings into the well-pretrained vocabulary manifold that minimizes interference with nearby points and accelerates the formation of a stable local structure for the new token. In contrast to prior randomized initialization strategies—whose target semantic position on the manifold is effectively unreachable under standard optimization—our method initializes each new token near the center of its target semantic region. This placement ensures that gradient-based training can efficiently refine the embedding toward the desired location, resulting in faster convergence and more reliable downstream performance.

---

### Note · Authors · 2026-01-06

I have read and agree with the venue's withdrawal policy on behalf of myself and my co-authors.